# Defining the Zero Dose Child: A Comparative Analysis of Two Approaches and Their Impact on Assessing the Zero Dose Burden and Vulnerability Profiles across 82 Low- and Middle-Income Countries

**DOI:** 10.3390/vaccines11101543

**Published:** 2023-09-28

**Authors:** Chizoba Wonodi, Brooke Amara Farrenkopf

**Affiliations:** International Vaccine Access Center, Johns Hopkins Bloomberg School of Public Health, Baltimore, MD 21231, USA; cwonodi1@jhu.edu

**Keywords:** zero dose children, zero dose definition, immunization equity, sociodemographic profile, indicators, vulnerability

## Abstract

While there is a coordinated effort around reaching zero dose children and closing existing equity gaps in immunization delivery, it is important that there is agreement and clarity around how ‘zero dose status’ is defined and what is gained and lost by using different indicators for zero dose status. There are two popular approaches used in research, program design, and advocacy to define zero dose status: one uses a single vaccine to serve as a proxy for zero dose status, while another uses a subset of vaccines to identify children who have missed all routine vaccines. We provide a global analysis utilizing the most recent publicly available DHS and MICS data from 2010 to 2020 to compare the number, proportion, and profile of children aged 12 to 23 months who are ‘penta-zero dose’ (have not received the pentavalent vaccine), ‘truly’ zero dose (have not received any dose of BCG, polio, pentavalent, or measles vaccines), and ‘misclassified’ zero dose children (those who are penta-zero dose but have received at least one other vaccine). Our analysis includes 194,829 observations from 82 low- and middle-income countries. Globally, 14.2% of children are penta-zero dose and 7.5% are truly zero dose, suggesting that 46.5% of penta-zero dose children have had at least one contact with the immunization system. While there are similarities in the profile of children that are penta-zero dose and truly zero dose, there are key differences between the proportion of key characteristics among truly zero dose and misclassified zero dose children, including access to maternal and child health services. By understanding the extent of the connection zero dose children may have with the health and immunization system and contrasting it with how much the use of a more feasible definition of zero dose may underestimate the level of vulnerability in the zero dose population, we provide insights that can help immunization programs design strategies that better target the most disadvantaged populations. If the vulnerability profiles of the truly zero dose children are qualitatively different from that of the penta-zero dose children, then failing to distinguish the truly zero dose populations, and how to optimally reach them, may lead to the development of misguided or inefficient strategies for vaccinating the most disadvantaged population of children.

## 1. Introduction

Addressing the problem of zero dose children is a central concern of the global immunization community because zero dose children constitute a gap in population immunity against vaccine-preventable diseases, and are emblematic of the unequal access to essential health services faced by families across the world, especially in low- and middle-income countries [1]. The Immunization Agenda 2030 (IA2030) has as one of its impact goals a 50% reduction in the number of zero dose children in the world by 2030 [2,3]. Similarly, the Gavi 5.0 strategy declares reaching zero dose children and missed communities a core priority with equity as the organizing principle [4]. This implies that the zero dose agenda is driven primarily by the moral, social, and economic imperative to address structural inequities and reach the most vulnerable and marginalized in the population with health services.

To drive efforts on the zero dose agenda, Gavi, the Vaccine Alliance has published the IRMMA framework (identify, reach, monitor, measure, and advocate) as an operational framework with which the Expanded Program on Immunization (EPI) programs of various countries can anchor tailored interventions [5]. This framework posits that in order to reach zero dose children and link them to services, EPI programs must first identify them [5]. However, in order to identify zero dose children, programs must first define them. So who is a zero dose child?

There is much discussion about the definition used to characterize zero dose children with questions about what antigen forms the basis of the definition and what reference age range should be considered [6,7]. From a purist perspective, the zero dose status should be assigned when a child has received no antigen from the immunization program. Estimating this purist definition of zero dose requires survey data, such as from Demographic and Health Surveys (DHSs) or Multi-Indicator Cluster Surveys (MICSs) data, or vaccination registry data that captures vaccination information at the individual level. In many low- and middle-income countries with no or incomplete vaccination registries, this analysis can only be done when survey data become available, usually at intervals of approximately five years [8,9]. Furthermore, these survey data do not provide estimates below the first sub-national level, precluding real-time monitoring at lower levels and greatly hindering performance management at the sub-district levels where it matters the most [10]. While survey data is limited in the frequency and granularity, it has the advantage of providing information on socio-demographic characteristics and thus helps improve our understanding of who is truly zero dose and why they are zero dose [10,11].

Given the limitations of the purist definition of a zero dose child, for operational purposes, zero dose has been defined by immunization experts as non-receipt of the first dose of the diphtheria-tetanus-pertussis-containing vaccine (DTP1), i.e., penta-zero dose [5,6]. This is based on practical considerations that administrative data, which is available in real time and down to the facility level, can be used to estimate the number of children who have missed out on essential vaccination services in a given locality and thus be useful for the regular and localized monitoring of immunization program performance on zero dose [12,13]. However, quality issues with routine administrative data, such as incomplete or missing reporting, inconsistency in reporting formats and denominator estimation challenges, are a common drawback of administrative data [12]. Faulty denominators may lead to either an overestimation or underestimation of the number of zero dose children. Sometimes this underestimation produces improbable values of negative zero dose children [8,14,15,16]. The lower the level and the smaller the geographic area, the more distorted the estimates from faulty administrative data become [15].

Using administrative data for immunization coverage rates can be problematic [17]. The numerator data may suffer from incompleteness due to non-reporting or delayed reporting [18]. Numerator data might be subject to intentional or unintentional inflation. Furthermore, a discrepancy may arise when individuals receive vaccinations in a different district from their residential district; they contribute to the numerators in the vaccinating district while they remain in the denominator of their residential district. Such a mismatch affects the quality of the coverage estimates.

On the denominator side, challenges stem from incomplete birth registration, the migration of individuals, and the use of outdated census data. These factors can adversely affect the accuracy of the population denominator used to calculate immunization coverage rates.

Although surveys are considered the gold standard, they are not without downsides. Sampling bias or the underrepresentation of some groups can occur in immunization coverage surveys. For example, certain groups, such as security-challenged individuals, might be difficult to access, leading to their limited inclusion in the survey sample [10]. Another factor that can contribute to bias is the inaccurate recall of vaccination history, particularly when mothers do not have a vaccination card to reference [19].

Moreover, discrepancies might arise due to administrative borders, where vaccination data might not align seamlessly across different administrative divisions. Additionally, there can be a social desirability bias among caregivers, where they may feel compelled to provide responses that are socially acceptable or desirable, potentially influencing the accuracy of reported vaccination rates [20].

By design, the practical definition does not account for vaccine doses that may be recommended at birth, such as BCG, Hepatitis B 0, and polio 0 (although recommendations differ across countries). There is also an implicit assumption that penta-zero dose children aged 12 months and above, who did not receive DTP1, also did not receive the antigens co-administered on the DTP schedule, or shots given at nine months and the second year of life (measles, DTP booster doses, and, in some countries, yellow fever or Japanese encephalitis vaccines). Choosing to ignore that some penta-zero dose children may have also received birth dose vaccines, and other vaccines, means that we are also ignoring the possibility that penta-zero dose children may have a different profile from children who are truly zero dose for all antigens.

Given the utility of the operational definition of zero dose as no-DTP1, it is important to understand how this definition compares to a definition of zero dose status that incorporates multiple vaccines to identify children who are truly unvaccinated. This analysis aims to understand the pros—in terms of ease of measurement and monitoring—and cons—in terms of an accurate count and sociodemographic profiling—when using each definition. We aim to understand how many more children are considered zero dose when utilizing the definition that includes only DTP1 (compared to multiple vaccines), whether the sociodemographic characteristics of zero dose children varies based on the definition used, and the number of vaccines and antigens received by children who have not received DPT1 but have been vaccinated.

## 2. Materials and Methods

### 2.1. Data

Data from the United States Agency for International Development’s (USAID) DHS and the United Nations Children’s Fund’s (UNICEF) MICS were utilized in our analysis [21,22]. The most recent survey with immunization data available from 2010 to 2020 from low- and middle-income countries were included. The information utilized in the DHS/MICS was collected through in-person interviews with women between 15 and 49 years of age on the health and immunization status of their children aged 12 to 35 months [23]. DHSs and MICSs are nationally representative, conducted approximately every five years, and follow a complex, two-stage cluster sampling design [23]. DHSs and MICSs are designed to be utilized together well, but there are a few notable differences [24]. For the DHS, only biological mothers are interviewed for information about children under five years, but the MICS includes biological mothers and primary caregivers for children under five living in the household [24]. In addition, the reference periods for some indicators on maternal care (i.e., antenatal care) are different across DHSs and MICSs, but will not affect or bias our analysis because the sample in our analysis is under two years and both surveys measure these indicators among children up to at least two years [24]. Both the DHS and MICS use the same approach and set of questions to ascertain child vaccination status, i.e., through information on the vaccine card or shared via the mother’s/caregiver’s report [25,26,27]. In addition, they both use similar approaches to ascertain whether the reported vaccine doses were received via immunization campaigns [25,26,27].

Data from the World Bank Group on the per capita gross national income of the year in which the survey was conducted were utilized to classify countries as low, lower-middle, and upper-middle income [28]. In addition, birth cohort estimates from the United Nations World Population Prospects 2022 were also utilized, as described below [29,30].

### 2.2. Country Selection

Low, lower-middle, and upper-middle-income countries with DHS/MICS data available from 2010 to 2020 were included in the analyses [21,22].

### 2.3. Outcome Measures

Children ages 12 to 23 months are included in this analysis, as they should have received at least the first dose of routine immunizations within their first year of life [31]. This age group is in line with other recent work and analyses, and allows for the comparison across DHS/MICS, WHO, and UNICEF Estimates of National Immunization Coverage (WUENIC) data, which consider the same age groups [7,29,32]. Vaccine receipt is confirmed either by vaccine card or record or caregiver recall when a vaccine record was not available in the DHS/MICS interview; including both records and caregiver recall gives a more conservative estimate of zero dose vaccination among households with missing vaccine cards [23]. We considered vaccines that were administered through any delivery platforms (i.e., via routine immunizations, campaigns, or child health days/weeks), as these were reported on the vaccine card and via caregiver recall [25,26,27]. The following three outcomes are utilized in our analysis: **Penta-zero dose**—Children 12 to 23 months that have not received the first dose of the diphtheria-tetanus-pertussis-containing vaccine, per DHS/MICS data. The pentavalent vaccine is commonly used in low- and middle-income countries as the vaccine to protect against diphtheria, tetanus, and pertussis, as it protects against those three diseases, hepatitis B and *Haemophilus influenzae* type b, and has been widely supported in the countries included in the analysis through Gavi support. The pentavalent vaccine replaced the DTP vaccine in many countries, hence the common usage of no-DTP1 to account for places where the pentavalent vaccine has not been introduced. The vaccine is given in three doses, usually at 6, 10, and 14 weeks of age. Oral polio, the pneumococcal conjugate vaccine, and the rotavirus vaccine (one dose at 14 week) are other antigens co-administered during the visits for pentavalent vaccines. If a child failed to get the first dose of the vaccine, it follows that they failed to get subsequent doses of pentavalent vaccine. It is assumed that they also failed to get the co-administered vaccines.**Truly zero dose**—Children 12 to 23 months that have not received any doses of each of the following vaccines, per DHS/MICS data: Bacille Calmette-Guérin (BCG), polio, pentavalent, and measles-containing vaccines (MCV). At least the first dose of each of these vaccines is recommended within the first year of life in most low- and middle-income countries [31]. As with the penta-zero dose definition, the assumption here is that having not received the pentavalent series, these children also did not receive other vaccines co-administered with the same schedule.**Misclassified zero dose children**—Children 12 to 23 months who have not received the first dose of the pentavalent vaccine but have received at least one dose of at least one of the following vaccines, per DHS/MICS data: BCG, polio, and measles-containing vaccines. This shows that this subset of the penta-zero dose children had indeed received some vaccination, but are ‘misclassified’ as being zero dose per the penta-zero dose definition.

### 2.4. Variables Selection

We reviewed the literature to identify sociodemographic factors related to poor access to immunization services [33]. In selecting from variables with data available in both the DHS and MICS, we included the following variables in our analyses: residence (rural vs. urban living), wealth quintile, maternal education level (no education vs. education (primary and above)), mother’s age group (adolescent (15–19 years) vs. adult (20–49 years)), location of delivery (home vs. facility), the level of antenatal care (ANC) visits (none (0 visits), low (1–3 visits), or 4+ visits), the number of maternal tetanus injections (none (0 times), low (1 time), or 2 or more), the sex of child (female/male), illness with diarrhea in the past two weeks (yes vs. no/caregiver does not know), illness with cough in the past two weeks (yes vs. no/caregiver does not know), illness with fever in the past two weeks (yes vs. no/caregiver does not know), treatment for recent cough/fever at a health facility (no vs. yes), and treatment for recent diarrhea at a health facility (no vs. yes).

### 2.5. Estimating the Zero Dose and Misclassified Populations

WUENIC data were utilized to ascertain the reported immunization target population for each country for the year of their included survey [29]. These data are provided by WUENIC and ascertained through the WHO and UNICEF Joint Reporting Form on immunization mechanism [34]. To estimate the number of penta-, truly, and misclassified zero dose children, the prevalence of each of these outcomes was multiplied by the target population in each country. This was then summed overall and by country income groups.

### 2.6. Conducting Statistical Analyses

Descriptive statistical analyses incorporating the complex survey design were utilized to generate the proportion of each zero dose outcome and the proportion of each characteristic across the full population and across the penta-, truly, and misclassified zero dose populations. The difference between proportions were tested between the truly zero dose and misclassified zero dose children utilizing adjusted Wald tests that accounted for the survey design [35]. Statistical significance was assessed at the alpha < 0.05 level. To generate standard errors required to conduct the adjusted Wald tests, strata with one sampling unit were centered at the grand mean, as this is a conservative approach to handle a single-unit stratum [36]. An analysis of missing data for each variable was conducted to explain the extent of missingness and potential biases. All statistical analyses were completed using Stata 14.2 software.

In addition, to understand the receipt of vaccines received by misclassified children, we created variables to consider the misclassified children who received one, two, or three other vaccines. We then assessed the singular vaccine or combination of vaccines received to generate frequency estimates by each grouping (i.e., percentage of misclassified zero dose children who received only polio vaccine).

### 2.7. Conducting Vulnerability Analyses

To identify the children that experience cross-cutting vulnerability and are at an elevated disadvantage compared to other children, we created two simple indicators. We did this to see how the understanding of zero dose status changed among the most vulnerable subset of the population. Indicator A considers overall vulnerability and includes variables related to household income, maternal education, and maternal health access. Among the children in the households in the poorest wealth quintile, we identified children with mothers who did not receive primary education and did not receive any antenatal care visits. Indicator B considers vulnerability related to limited access to maternal care services and includes children who were born at home and who have mothers who did not receive any ANC visits nor tetanus injections.
**Indicator A—Overall vulnerability**: Poorest wealth quintile, no maternal education, and 0 ANC visits**Indicator B—Health access vulnerability**: 0 ANC visits, 0 tetanus injections, and home delivery

## 3. Results

Eighty-two (82) low- and middle-income countries were included in our analysis, with 27 low-income, 37 lower-middle income, and 18 upper-middle-income countries (Appendix A). In total, 194,829 observations were included in the analysis, with 30.3% (N = 58,998) of the observations from low-income countries (LIC), 58.8% (N = 114,572) from lower-middle-income countries (LMIC), and 10.9% (N = 21,259) from upper-middle-income countries (UMIC). Overall, 47 DHSs and 35 MICSs were included. The median survey year was 2017, and 75% of the surveys were published in 2015 or later.

When defining zero dose as children aged 12–23 months who have not received a single dose of the pentavalent vaccine—called *‘penta-zero dose’* for this analysis—14.2% of children across the 82 countries in our analysis are classified as zero dose (Table 1). When defining zero dose as the children aged 12 to 23 months who have not received a single dose of the BCG, polio, pentavalent, or measles-containing vaccines (MCV)—called *‘truly zero dose’* for this analysis—7.5% of children are classified as zero dose. This means that 6.6% of children aged 12 to 23 months are *‘misclassified’ as zero dose*; that is, they have not received any doses of the pentavalent vaccine but have received at least one other vaccine (BCG, polio, or MCV) and have therefore had a connection with the immunization system.

Nearly half (46.5%) of the children considered zero dose using the penta-zero dose definition are misclassified as zero dose. This is similar to the proportion of children misclassified in low-income countries (LICs) and lower-middle-income countries (LMICs), where nearly one in five (19.1%) children are penta-zero dose in LICs, and more than 1 in 8 are penta-zero dose in LMICs. Although there is a smaller overall proportion of zero dose children in upper-middle-income countries (UMICs) (9.4% penta-zero dose and 4.0% truly zero dose), a greater proportion (57.6%) are misclassified, meaning that over half of the zero dose children are not truly zero dose and have successfully been reached at least once by the immunization system. (Please see Appendix A for the national prevalence estimates of penta-zero dose, truly zero dose, and misclassified zero dose children and details about the immunization schedule in each country).

When incorporating population estimates to ascertain the number of zero dose children aged 12 to 23 months across the countries in our analysis, there are approximately 13,741,120 penta-zero dose children and 7,328,670 ‘truly’ zero dose children (Table 2). The number of zero dose children is nearly double when using the penta-zero dose definition, with an estimated 6,412,450 children misclassified as zero dose but have been vaccinated with at least one antigen. This can have equity implications on programs that aim to vaccinate the most vulnerable children or aim to vaccinate a target proportion or number of ‘zero dose’ children. In upper-middle-income countries, there are a higher number of misclassified zero dose children than truly zero dose children, yet we see a higher number of truly zero dose children than misclassified children in LICs and LMICs. To note, the number of zero dose and misclassified zero dose children across each country’s income group is dependent on the number and population of the countries in each group included in the analysis, with a smaller number of UMICs included in this analysis.

Table 3 shows the proportion of factors potentially associated with non-immunization, in the full population in the analysis and among penta-zero dose children, truly zero dose children, and misclassified zero dose children, with *p*-values to test the difference between proportions in the truly zero dose and misclassified populations. Approximately two-thirds (65.3%) of all children in the analysis (regardless of vaccination status) live in rural areas, but three-fourths of penta-zero dose (74.8%) and truly zero dose (75.8%) children live in rural areas, with no significant difference in the percentage of children living in rural areas between truly zero dose and misclassified zero dose children.

Over one third (36%) of the penta-zero dose children are in the poorest wealth quintile, with 37.5% of the truly zero dose children in the poorest quintile. The ‘misclassified’ zero dose children are less likely to be in the lowest wealth quintile (34.3%), and the difference between the proportion of children in the lowest wealth quintile is significantly different in the truly zero dose and misclassified zero dose group (*p* = 0.005). Truly zero dose children are 9% more likely to be in the poorest wealth quintile than the misclassified zero dose children, with the differences statistically significant in LICs but not in LMICs or UMICs.

Urban vs. rural living, the sex of child, mother’s adolescent age, and maternal education are not significantly different between truly and misclassified zero dose children. We do, though, see a higher proportion of penta- and truly zero dose children living in rural areas, having mothers of adolescent age, and having no education than we see in the full population, suggesting that these are factors associated with zero dose status; however, this does not differ by definition used.

There is a higher percentage of children who have mothers without any ANC visits and without any tetanus injection among zero dose children (regardless of definition) than the full population, overall and in each country income group. We also see a difference between the truly and misclassified zero dose children for these characteristics, suggesting that a lack of maternal care is more common among zero dose children, and especially among truly zero dose children. A similar relationship is seen with home deliveries; there is a higher proportion of children born at home in the zero dose population (regardless of definition) than the full population, but there is also a higher proportion in truly zero dose children than misclassified zero dose children.

We also wanted to understand how child illness and care-seeking may differ across the population of children prioritized in each zero dose definition. This will enable us to understand whether the population of zero dose children under either definition has better connection to care, which can have programmatic implications of how zero dose children are targeted. Interestingly, children who are truly zero dose are less likely to have experienced recent diarrhea, cough, and fever overall but were also less likely to access treatment when ill (children with illness in the past two weeks were compared with children without illness/unknown illness in the past two weeks) (Table 4). While children who were truly zero dose were overall less likely to have diarrhea in the past two weeks than misclassified zero dose children (19.2% vs. 23.2% of children experienced diarrhea, respectively, *p* < 0.001), the truly zero dose children who were ill were less likely to access treatment at facilities for diarrhea if they were ill (46.6% of children with diarrhea) compared to the misclassified zero dose children (55.1%) (*p* = 0.001), suggesting that they face additional barriers in accessing vaccines and care; similar and statistically significant findings were also seen in LMICs. Similarly, overall and in LICs and LMICs, truly zero dose children were less likely to experience cough and/or fever in the past two weeks but were also less likely to have received care if they were ill with fever and/or cough (33% less likely than misclassified zero dose children in LICs, *p* = 0.005). These results were not observed in UMICs, where truly zero dose children were nearly twice as likely to receive care for cough and/or fever than misclassified zero dose children, suggesting that there may be other barriers to seeking or accessing vaccination services and/or care in UMICs that were not included in this analysis.

To understand the extent of zero dose status and misclassification among the most vulnerable subset of the population, we created simple indicators of cross-cutting vulnerability to identify households at an elevated vulnerability (Table 5). ‘Overall vulnerability’ is defined as children in the poorest wealth quintile and with mothers who have received no education and no ANC visits. ‘Health access vulnerability’ is defined as children with mothers who received no ANC visits and no tetanus injections and had a home delivery. While a small proportion (3.6%) of children in the overall population are classified as having ‘Overall Vulnerability’ in our analysis, a substantial proportion of the ‘overall vulnerable’ population are penta-zero dose (12.1%) and truly zero dose (13.6%), with a statistically significant difference of overall vulnerability between the truly zero dose and misclassified zero dose children (<0.001). This is concentrated in LICs and LMICs, with the strongest difference seen in LICs, suggesting that the misclassified zero dose children are less vulnerable than the truly zero dose children.

In the second indicator, which considers vulnerability related to limited access to maternal care services, although fewer than 1 in 20 children (4.9%) are considered ‘Health access vulnerable’ by this indicator; in contrast, 1 in 4 (25.2%) truly zero dose children and over 1 in 5 (21.3%) penta-zero dose children are ‘health access vulnerable’, a level considerably higher than that seen in the general population. Overall, in LICs and in LMICs, the proportion of truly zero dose children considered vulnerable by this indicator is significantly different than that of misclassified zero dose children, again showing less vulnerability in those who are misclassified as zero dose. A very low percentage of the population in UMICs was vulnerable in either of these indicators, as households in UMICs are more connected to the services included in the indicators, which suggests that this may not be an appropriate approach to detect cross-cutting vulnerability in zero dose children in UMICs. Moreover, it would be useful to consider demand-side drivers of vaccine uptake to understand if zero dose status in UMICs is driven more by children who are hard to vaccinate (demand issues) than by children who are hard to reach (access issues) [37].

Overall, truly zero dose children are more likely to be considered in our vulnerability indicators than misclassified children, but the magnitude of these effect sizes differs in LICs and LMICs, suggesting the truly zero dose children have different vulnerabilities in LICs vs. LMICs. In LICs, truly zero dose children are 49% more likely to be considered ‘overall vulnerable’ and 25% more likely to be considered ‘health access vulnerable’ compared to misclassified zero dose children. This suggests that in LICS, the truly zero dose children have more multi-level vulnerability (i.e., education and poverty) than those misclassified. In comparison, in LMICs, truly zero dose children are 22% more likely to be considered ‘overall vulnerable’ and 57% more likely to be considered ‘health access vulnerable’ compared to misclassified zero dose children. This means that the truly zero dose children are considerably more vulnerable in terms of their access to health services than misclassified children, and that using the penta-zero dose definition may dilute the recognition of the health access vulnerability that confronts the most marginalized sub-set of zero dose children in LMICs.

Overall, 49.7% of children classified as penta-zero dose have received at least the first dose of exactly one other vaccine (BCG, polio, MCV), with half (50.3%) of the misclassified zero dose children have been contacted by the immunization system at least twice (Table 6). This is similar in LICs and LMICs, but a higher proportion of penta-zero dose of children in UMICs have received at least two vaccines (59.9%), with a quarter of penta-zero dose children having receiving BCG, polio, and measles-containing vaccines in UMICs.

Of the misclassified children who have received one vaccine, 58.1% of them have received the polio vaccine, meaning that they were only reached by the polio vaccine. In LICs, this was higher: 68.0% of the misclassified zero dose children had only received the polio vaccine (Table 7). Notably, in UMICs, 88.9% of misclassified zero dose children who received only one vaccine had only received the BCG vaccine. Less than 5% of children overall and in each country’s income groups received only the measles vaccine, which could in part be a result of the measles vaccine being recommended later in the first year than the other vaccines. Of the children who received two vaccines, most (72.6%) received both polio and BCG vaccines.

## 4. Discussion

This analysis set out to investigate how using two different approaches to define zero dose children changes the count and profile of zero dose children overall and by country income level grouping. We found that the penta-zero dose definition, the widely accepted operational definition, which is defined based on the non-receipt of the pentavalent vaccine, overestimates the number and proportion of children who are truly zero dose—that is, children who have received no single vaccine dose from the immunization system, as measured by the non-receipt of BCG, polio, pentavalent and measles vaccines. Only about half of the children defined as penta-zero dose were truly zero dose. The other half were misclassified zero dose children though they had received one or more vaccines, indicating that these children were reached at some point by the vaccines system, although they remained under-vaccinated and vulnerable to vaccine-preventable diseases.

The demographic correlates of being penta-zero dose are well established [6]. Some of these, such as living in the rural area, living in poorer households, having mothers of adolescent age, or having mothers with no education, do not differ by the zero dose definition used. However, some differences were observed. Truly zero dose children are 9% more likely to be in the poorest wealth quintile than the misclassified zero dose children, with the differences being statistically significant in LICs and not in the other two country income groups.

In this analysis, we present two novel vulnerability measures derived from variables available in the DHS dataset. The first measure assesses overall vulnerability, which combines socio-demographic risk factors with barriers to healthcare access to estimate a multidimensional burden of vulnerability. The second measure specifically focuses on health access vulnerabilities.

Our findings reveal distinct vulnerability patterns when comparing truly zero dose children to misclassified zero dose children, based on country income groupings. In low-income countries (LICs), truly zero dose children are 49% more likely to be categorized as ‘overall vulnerable’ compared to misclassified zero dose children. Conversely, in lower-middle-income countries (LMICs), truly zero dose children experience a 57% higher burden of ‘health access vulnerability’—or lack of access to ANC, tetanus injections, and facility deliveries—compared to misclassified zero dose children. This suggests that in LICs, truly zero dose children are primarily distinguished by multidimensional barriers, while in LMICs, they are predominantly differentiated by health access vulnerabilities.

Recognizing that the vulnerability of truly zero dose children, both in terms of overall vulnerability and access to healthcare, may be obscured when using the penta-zero dose definition highlights the need for heightened attention to these vulnerabilities during program planning. This is particularly true in settings where resources are constrained and programs are able to target interventions. However, implementing this approach in practice poses challenges due to data limitations, which make it difficult to accurately distinguish children who are truly zero dose from those who are not.

The main concern with misclassifying children as zero dose is the implied dilution of attention to the children who are truly zero dose, who have been completely missed by the health system. Currently, the zero dose agenda commands significant resolve, resources, and effort from Gavi, the Vaccine Alliance partners, and other global, regional, and country immunization partners [2,4]. These efforts are geared toward addressing immunization equity gaps in the last mile by putting the last child first [2,4,38]. Under this framing, truly zero dose children should be given first priority.

But how much does it matter if the use of penta-zero dose means that programs are unable to distinguish truly zero dose children from misclassified zero dose children when the goal is to close immunity gaps amongst the children at the highest risk of infection? Are misclassified zero dose children really at a relatively lower risk of vaccine-preventable diseases than truly zero dose children? It is unlikely to be the case. Among penta-zero dose children who received only one other vaccine, two thirds (68%) had received the polio vaccine, while only 4.6% had received the measles vaccine. This means that the majority were unprotected against measles, which is a major cause of disease outbreaks and the tracer disease for the immunization system. Since the vast majority of misclassified zero dose children were not vaccinated against measles, it could be argued that not much is lost by focusing on penta-zero dose children since most of them are zero dose for the measles vaccine.

### 4.1. Strengths of the Analysis

A strength of the analysis is that it includes data from 82 low- and middle-income countries, which allows for the comprehensive, global understanding of zero dose populations, with results available overall and at the country income level. The nature of this multi-country analysis and the use of publicly available secondary data allows for comparisons across different populations. Utilizing multiple nationally representative surveys enabled us to describe the characteristics and connection with the health system of over 13 million children, as represented by the nearly 200,000 observations in our analysis. In addition, the large sample size enabled us to run several analyses on sub-populations (i.e., the most vulnerable 5% of the population) and still be able to make statistical inferences at the alpha < 0.05 level. While this analysis brings the new insight comparing the definitions and introducing the concept of misclassified zero dose children, it aligns with sociodemographic factors that are considered in other analyses, which can further allow for comparison and alignment across research, and focuses on factors that are specific to zero dose status to better understand these definitions [39,40,41,42]. It also looks at the zero dose definitions and the extent of misclassification in subsets of the population that is at an elevated vulnerability. In addition, the timeliness of the analysis is important, as much action and research is ongoing in the zero dose space. The findings from this analysis can be replicated as additional DHSs, MICSs, and other data sources are made available.

### 4.2. Limitations of the Analysis

We were limited by the available data in our analysis, so several low- and middle-income countries without DHS/MICS data available since 2010 were not included. In addition, a small number of UMICs were included in our analysis due to data availability, which hinders the generalizability of these results to other UMIC settings. DHSs/MICSs are reported to undersample the population in urban poor settings, as DHSs/MICSs rely on census estimates that can be less reliable in urban areas [43,44]. In addition, due to security challenges, surveys may not have access to areas experiencing conflict, which can undersample children in conflict or humanitarian settings [45]. Both urban poor and conflict-affected settings are identified as areas with a high proportion of zero dose children by the Equity Reference Group and has been widely accepted in the global immunization community, so additional resources and analyses are likely needed to understand the zero dose population and immunization dynamics in these settings [38,45,46]. While we attempted to include the relevant variables for zero dose status, there may be variables not included in our analysis. In addition, confounding effects may be present in the variables related to recent child illness and care-seeking, as vaccination is on the causal pathway to child illness and severity.

To ascertain vaccine status, we included both vaccine card and caregiver recall data to limit bias in vaccine coverage estimates and to avoid overestimating zero dose status among vaccinated children without vaccine cards [20]. Yet, there are varying levels of validity and reliability in caregiver recall, and caregiver recall has the potential for recall and social desirability biases, which would underestimate zero dose status [19,20,47,48]. Since we did not account for whether information was from vaccine records or caregiver recall in our analyses, we are unable to assess how this may differ across zero dose definitions and across contexts, where practices around availability and retention of vaccine cards/records may differ [49]. Our analysis looked at the misclassification of vaccinated children as zero dose when using the penta-zero dose definition based on the responses reported in the DHSs/MICSs. We did not assess the quality of the immunization data in the DHS and MICS (i.e., whether a child with reported DTP1 vaccination was actually vaccinated), which could bias our analyses if the data quality is poor.

While the profile of zero dose children in LICs and LMICs was generally similar, the profile at times differed in UMICs (i.e., in treating child illness and in the vaccines received by misclassified zero dose children), suggesting that factors not included in our analysis may impact non-immunization more in those settings, including demand-related factors. The scope of analysis was to only consider household-level factors, not health systems factors or demand-related factors, which likely impact zero dose status as well.

Lastly, all data included were collected prior to the COVID-19 pandemic, they do not capture the changing immunization and health landscape. UNICEF, WHO, and other researchers have reported that immunization coverage estimates have declined since the COVID-19 pandemic, so our estimates likely underrepresent the number and proportion of zero dose children in the countries included in our analysis [46,50]. Although the data is somewhat older for some of our countries, we find that it is appropriate to include, as a major driver of the focus on zero dose children is the stagnating immunization coverage rates over the past decade [29].

In addition, we chose not to disaggregate immunization data by delivery platform (i.e., routine immunization, campaigns, or child health days), so that we could obtain an understanding of the reach of vaccination activities and identify those missed through all delivery platforms. Future analyses, though, could show how our understanding of zero dose status—including the sociodemographic profile and number of zero dose children—may differ when campaign vs. routine immunizations are used to define zero dose status, as this information could inform program design.

## 5. Conclusions

Questions regarding the two definitions of zero dose children have been thoroughly examined in our analysis. We have found that the purist definition of truly zero dose can identify a subgroup of penta-zero dose children with a higher vulnerability profile than the other subgroup, which we call the misclassified zero dose children. This differentiation proves valuable in understanding the varying levels of vulnerability among zero dose children.

We also found the penta-zero dose construction provides a pragmatic definition of zero dose that can be easily measured using existing routine data. This makes it a practical definition for routine program planning, monitoring, and data utilization for action. In contrast, the truly zero dose definition relies on survey data available only at multi-year intervals, rendering it impractical for ongoing program monitoring.

Finally, it is worth noting that although the truly zero dose population demonstrates a higher vulnerability profile compared to misclassified zero dose children, both groups share common characteristics, in particular, most are zero dose for measles vaccination. This indicates that the penta-zero dose definition not only offers easier measurement, but also captures the children with the most critical immunity gaps, making it a valuable definition for identifying and addressing immunization program needs to achieve vaccine equity.

## Figures and Tables

**Table 1 vaccines-11-01543-t001:** Proportion of children aged 12 to 23 months in 82 low- and middle-income countries who are penta-zero dose, truly zero dose, and misclassified as zero dose, overall and by country income level using DHS/MICS data.

	Penta-Zero Dose Children	Truly Zero Dose Children	Misclassified Zero Dose Children	Percentage of Penta-Zero Dose Children that Are Misclassified as Zero Dose
**Percentage of zero dose children,** overall	14.2	7.5	6.6	46.5
Low-income countries (LICs)	19.1	10.4	8.6	45.0
Lower-middle-income countries (LMICs)	13.0	7.0	6.1	46.9
Upper-middle-income countries (UMICs)	9.4	4.0	5.4	57.4

**Table 2 vaccines-11-01543-t002:** Using population data to estimate the number of children aged 12 to 23 months in 82 low- and middle-income countries who are penta-zero dose, truly zero dose, and misclassified as zero dose, overall and by country income level.

	Penta-Zero Dose Children	Truly Zero Dose Children	Misclassified Zero Dose Children
**Number of children,** overall	13,741,120	7,328,670	6,412,450
LIC	4,066,966	2,242,605	1,824,360
LMIC	9,075,447	4,828,183	4,247,264
UMIC	598,708	257,882	340,826

**Table 3 vaccines-11-01543-t003:** Characteristics of children overall and in each zero dose definition, with tests to detect difference between proportions in truly zero dose and misclassified zero dose children.

			Comparison between Proportions			
	Full Population *(Regardless of Vaccination Status)*	Penta-Zero Dose Children	Truly Zero Dose Children	Misclassified Zero Dose Children	Comparison of Proportions *(Proportion of Characteristic in Truly* vs. *Misclassified Zero Dose Children)*	*p*-Value *(Comparing Truly Zero Dose and Misclassified)*
**N *,** overall	194,829	29,155	15,966	13,189	-	-
LIC	58,998	11,500	6418	5082	-	-
LMIC	114,572	15,289	8771	6518	-	-
UMIC	21,259	2366	777	1589	-	-
**Proportion rural,** overall	65.3	74.8	75.8	73.6	1.03	0.054
LIC	73.9	82.6	84.1	80.9	1.04	0.055
LMIC	65.5	74.3	74.5	74.0	1.01	0.747
UMIC	34.6	29.5	28.1	30.5	0.92	0.613
**Proportion in the poorest wealth quintile,** overall	22.8	36.0	**37.5**	**34.3**	**1.09**	**0.005**
LIC	22.4	32.8	**35.9**	**29.0**	**1.24**	**<0.001**
LMIC	22.8	38.1	38.8	37.4	1.04	0.329
UMIC	23.5	27.2	27.7	26.8	1.03	0.890
**Child gender, proportion female,** overall	48.9	49.5	49.0	50.1	0.98	0.355
LIC	50.1	50.4	51.4	49.3	1.04	0.303
LMIC	48.4	49.0	47.8	50.4	0.95	0.076
UMIC	50.1	51.0	50.7	51.2	0.99	0.943
**Adolescent mother, proportion with mother aged 15–19 years,** overall	6.1	7.4	7.5	7.4	1.01	0.887
LIC	8.2	9.4	9.5	9.4	1.01	0.923
LMIC	5.2	6.4	6.5	6.2	1.05	0.615
UMIC	8.5	9.8	7.9	11.1	0.71	0.275
Mother without education, proportion with mother who did not receive primary education, overall	27.5	51.0	51.5	50.5	1.02	0.443
LIC	40.8	54.3	56.2	51.9	1.08	0.057
LMIC	25.5	52.2	51.5	52.9	0.97	0.396
UMIC	4.7	12.4	**8.8**	**15.1**	**0.58**	**0.017**
** *Maternal health access* **						
**Proportion with mothers with no ANC visits,** overall	11.6	33.5	**39.9**	**26.5**	**1.51**	**<0.001**
LIC	12.3	33.2	**38.6**	**27.3**	**1.41**	**<0.001**
LMIC	12.3	35.5	**42.2**	**28.0**	**1.51**	**<0.001**
UMIC	1.6	3.5	5.1	2.5	2.04	0.101
**Proportion with mothers with low ANC visits (1–3 visits),** overall	28.6	29.6	**26.6**	**32.9**	**0.81**	**<0.001**
LIC	40.2	36.7	35.0	38.5	0.91	0.106
LMIC	26.7	27.4	**23.5**	**31.9**	**0.74**	**<0.001**
UMIC	9.1	16.8	17.7	16.3	1.09	0.761
**Proportion with mothers with no tetanus injections,** overall	17.5	42.1	**46.9**	**37.0**	**1.27**	**<0.001**
LIC	22.7	42.6	**47.2**	**37.6**	**1.26**	**<0.001**
LMIC	16.4	43.4	**48.0**	**38.3**	**1.25**	**<0.001**
UMIC	10.6	12.7	14.4	11.4	1.26	0.458
**Proportion with mothers with low tetanus injections (1 injection),** overall	20.1	17.1	**15.5**	**18.7**	**0.83**	**0.001**
LIC	23.7	21.1	**18.9**	**23.3**	**0.81**	**0.008**
LMIC	17.7	14.6	**13.2**	**16.2**	**0.81**	**0.004**
UMIC	35.9	34.4	41.8	29.2	1.43	0.178
**Proportion delivered at home,** overall	27.4	54.5	**58.5**	**49.9**	**1.17**	**<0.001**
LIC	36.0	58.0	**60.7**	**54.8**	**1.11**	**0.007**
LMIC	26.7	55.7	**59.9**	**50.9**	**1.18**	**<0.001**
UMIC	5.7	11.4	11.1	11.5	0.97	0.856

Bolded results are statistically significant at alpha < 0.05. * N represents the unweighted number of observations for each sub-population in the dataset. Appendix B provides a missingness analysis for the variables included in this analysis.

**Table 4 vaccines-11-01543-t004:** Characteristics of recent child illness and treatment access when ill, overall and in each zero dose definition, with tests to detect difference between proportions in truly zero dose and misclassified zero dose children.

			Comparison between Proportions			
	Full Population *(Regardless of Vaccination Status)*	Penta-Zero Dose Children	Truly Zero dose Children	Misclassified Zero dose Children	Comparison of Proportions *(Proportion of Characteristic in Truly* vs. *Misclassified Zero Dose Children)*	*p*-Value *(Comparing Truly Zero Dose and Misclassified)*
**Proportion with diarrhea in past two weeks,** overall	19.3	21.1	**19.2**	**23.2**	**0.83**	**<0.001**
LIC	24.5	22.5	20.2	25.3	0.80	0.202
LMIC	17.9	20.8	**19.0**	**22.7**	**0.84**	**0.001**
UMIC	16.2	15.1	11.9	17.5	0.68	0.132
**Proportion with cough in past two weeks,** overall	25.4	22.0	**19.2**	**25.2**	**0.76**	**<0.001**
LIC	26.3	23.2	**21.3**	**25.4**	**0.84**	**0.014**
LMIC	25.0	21.5	**18.3**	**25.0**	**0.73**	**<0.001**
UMIC	26.4	22.5	17.6	25.9	0.68	0.164
**Proportion with fever in past two weeks,** overall	25.9	26.8	**23.8**	**30.1**	**0.79**	**<0.001**
LIC	27.2	27.1	**23.5**	**31.4**	**0.75**	**<0.001**
LMIC	26.0	27.2	**24.7**	**30.2**	**0.82**	**<0.001**
UMIC	20.1	17.0	**10.4**	**21.6**	**0.48**	**0.001**
** *Received treatment at facilities* **						
**Children with diarrhea in past two weeks**						
**N *,** global	37,936	14,738	20,447	2751	-	-
**Proportion who received treatment,** overall	57.6	46.9	**43.3**	**50.1**	**0.86**	**0.004**
LIC	47.1	32.4	29.4	35.2	0.84	0.143
LMIC	62.7	54.6	**50.7**	**58.3**	**0.87**	**0.006**
UMIC	46.4	36.7	32.5	38.7	0.84	0.585
**Children with cough and/or fever in past two weeks**						
**N *,** overall	59,704	20,596	33,543	5565	-	-
**Proportion who received treatment,** overall	60.2	51.0	**46.6**	**55.1**	**0.85**	**<0.001**
LIC	45.5	34.2	**29.3**	**38.9**	**0.75**	**0.005**
LMIC	65.4	59.5	**54.6**	**64.0**	**0.85**	**<0.001**
UMIC	58.5	53.1	**76.1**	**41.0**	**1.86**	**<0.001**

Bolded results are statistically significant at alpha < 0.05. * N represents the unweighted number of observations for each sub-population in the dataset. Note: Data were not collected on recent cough, diarrhea, and fever for Kazakhstan, Serbia, Thailand, and Tunisia, so these countries were excluded for the analysis of recent illness and treatment if ill. In addition, data on recent fever were not collected in El Salvador, Sudan, and Vietnam, so these countries were excluded for the analysis of ‘Fever in past two weeks’; these countries were included in the analysis of ‘Children with cough and/or fever in past two weeks who sought treatment’ to consider the children who sought treatment when ill with cough. A data missingness analysis is included in Appendix B.

**Table 5 vaccines-11-01543-t005:** Understanding the zero dose population by definition among the most vulnerable households.

			Comparison between Proportions			
	Full Population *(Regardless of Vaccination Status)*	Penta-Zero Dose Children	Truly Zero Dose Children	Misclassified Zero Dose Children	Comparison of Proportions *(Proportion of Characteristic in Truly* vs. *Misclassified Zero Dose Children)*	*p*-Value *(Comparing Truly Zero Dose and Misclassified)*
**Indicator A—Overall vulnerability:** Poorest wealth quintile, no maternal education, 0 ANC visits						
**Proportion classified as vulnerable (overall),** overall	3.6	12.1	**13.6**	**10.5**	**1.30**	**<0.001**
LIC	3.1	8.8	**10.3**	**6.9**	**1.49**	**0.004**
LMIC	4.1	14.5	**15.8**	**13.0**	**1.22**	**0.010**
UMIC	0.0	0.1	0.2	0.0	n/a	0.105
**Indicator B—Health access vulnerability:** 0 ANC visits, 0 tetanus injections, home delivery						
**Proportion classified as vulnerable (no maternal care),** overall	4.9	21.3	**25.2**	**17.0**	**1.48**	**<0.001**
LIC	7.7	22.1	**24.0**	**19.7**	**1.25**	**0.012**
LMIC	4.5	22.4	**27.0**	**17.2**	**1.57**	**<0.001**
UMIC	0.2	0.9	1.6	0.4	4.00	0.085

Bolded results are statistically significant at alpha < 0.05.

**Table 6 vaccines-11-01543-t006:** Number of vaccinations received among misclassified zero dose children.

Number of Vaccines Received among Misclassified Zero dose Children	Misclassified Zero Dose Children, Overall	Misclassified Zero Dose Children in Low-Income Countries	Misclassified Zero Dose Children in Lower-Middle-Income Countries	Misclassified Zero Dose Children in Upper-Middle-Income Countries
**N ***	13,189	5082	6518	1589
1 vaccine (%)	49.7	47.0	51.8	40.2
2 vaccines (%)	33.1	35.5	31.8	34.7
3 vaccines (%)	17.2	17.5	16.4	25.2

Column totals may not equal 100 due to rounding. * N represents the unweighted number of observations for each sub-population in the dataset.

**Table 7 vaccines-11-01543-t007:** History of vaccine(s) received among misclassified zero dose children.

	Misclassified Zero Dose Children with 1 Vaccine (%)	Misclassified Zero Dose Children with 2 Vaccines (%)
**Polio,** overall	58.1	-
LIC	68.0	-
LMIC	64.9	-
UMIC	6.7	-
**BCG,** overall	37.3	-
LIC	27.5	-
LMIC	30.5	-
UMIC	88.9	-
**MCV,** overall	4.6	-
LIC	4.5	-
LMIC	4.6	-
UMIC	4.5	-
**Polio + BCG,** overall	-	72.6
LIC	-	75.1
LMIC	-	70.9
UMIC	-	77.8
**Polio + MCV,** overall	-	16.2
LIC	-	17.9
LMIC	-	16.6
UMIC	-	3.1
**BCG + MCV,** overall	-	11.2
LIC	-	7.0
LMIC	-	12.5
UMIC	-	19.0

Note: Please see Appendix A for country-specific details on national vaccine schedules to aid with interpretation of possible vaccines received.

## Data Availability

All data are publicly available for download from the Demographic and Health Survey Program (https://dhsprogram.com, accessed on 14 April 2021) and the Multiple Indicator Cluster Survey program (https://mics.unicef.org/surveys, accessed on 14 April 2021).

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
