# Peer review of "Defining the Zero Dose Child: A Comparative Analysis of Two Approaches and Their Impact on Assessing the Zero Dose Burden and Vulnerability Profiles across 82 Low- and Middle-Income Countries"

_vaccines, 2023, doi:10.3390/vaccines11101543_

Round 1
Reviewer 1 Report
This paper presents an analysis comparing the definitions of "zero-dose" (ZD) and some implications in terms of magnitudes and characteristics. They compare the ZD definition adopted by the Immunization Agenda 2030 (IA2030) and Gavi 5.0, or what the authors refer to as “penta zero-dose” – and that I strongly recommend changing to DTP zero dose (see later) – i.e., children that by 12 months have not received any dose of a diphtheria-tetanus-pertussis (DTP)- containing vaccine [calculated as 1 minus the proportion vaccinated with DTP1] with another more strict definition based on no evidence of any vaccination in survey data.
Major comments
There is an implicit message that the 82 countries in the analysis represent the whole world (the words “global/globally” are used), or at least all low-and middle income countries (LMIC). There are 136 countries in the LMIC category, as per the 2023 World Bank classification, (or 133 as Venezuela and two small island countries remain un-classified) and there is no list of the 82 included countries, but knowing the DHS program there is likely an overrepresentation of Africa. Thus, I believe the paper should avoid over generalizing the findings as if the sample of included countries truly represents the whole world or all LMICs.
Recommendations related to the above
· Change “penta zero dose” to DTP/DPT zero dose”. Not every LMIC uses “penta” or DTP-Hib-HepB and in near future many countries may transition to an hexavalent including IPV. Plus the definition endorsed by IA2030 is really zero DTP-containing vaccine.
· Replace “globally/global” with the word “overall” or equivalent, throughout
· Include a table or supplemental material listing the included countries/surveys; a map could also help better grasp the distribution.
· If a table is added (highly recommended from my perspective), I’d suggest including country name, income classification, survey year, and schedule, as not every country recommends polio 0 or measles-containing vaccine at 9 months.
o For example, if the Suriname DHS was used, that country doesn’t recommend BCG, nor polio 0 and has MMR in the schedule at age 1 year.
· Related to the previous point, a concern I have is that it is unclear if the analysis accounted for specificities in recommended schedules at time of the DHS.
o Again, it would not make sense to include a country like Suriname in the BCG analysis
· We don’t know what proportion of the data comes from recall and what comes from documented evidence (card or facility record). A table with all surveys included could also have this information.
· The issue of misclassification must be considered and at least discussed. Children who are classified as “truly zero dose” are unlikely to have cards whereas DTP zero dose, may or may not have a card; misclassifications are likely either way. DHS and MICS can ascertain differently vaccination in the probing questions, leading to differential inclusion, or not, of campaign doses.
Other comments by section
Introduction
Cite the IA2030 M&E framework
· Lines 80-86 mentions some issues with admin data. This could be further explained. For a broader discussion on additional issues like mismatch between people where live and where are vaccinated see report from Data WG to SAGE in Oct 2019, summarized by Scobie et al., 2020 https://pubmed.ncbi.nlm.nih.gov/32950304/ . Rau et al. 2022 https://pubmed.ncbi.nlm.nih.gov/36962284/ explores DQ by country.
· To be more balanced, I think issues with surveys should also be described, and worse, mentioned how is also more likely to have issues with surveys in places with “bad” admin data (see Cutts et al. 2016 https://pubmed.ncbi.nlm.nih.gov/27349841/ and reply to a letter the same year https://pubmed.ncbi.nlm.nih.gov/27899197/ )
· Lines 87-88, to be more accurate may want to say “recommended” rather than “given” at birth, and qualify the statement by indicating that this is in many countries, but not universally
· Line 91, may also mention the DTP boosters and Japanese Encephalitis to make paper more descriptive of various parts of the world
Materials and Methods
· Clarify what was done if >1 survey in the study period
· Provide more information on what was used and countries included, in the form of a table for example, as described early.
· Explain better differences between MICS and DHS, like DHS only including biological children of women in the women questionnaire vs. MICS asking about all children; differences in that only one of these surveys tries to ascertain maternal tetanus vaccination by card, while the other is only recall
· Explain better the flow of vaccination status ascertainment, and how the different surveys try to handle/exclude campaign doses.
· Line 108, “approximately every five years” this seems about correct but earlier a more frequent cadence was implied/mentioned
· Lines 118-119. The second part of the sentence “Children ages 12 to 23 months are included in this analysis, as they should have received routine immunizations within their first year of life” is misleading as routine immunizations include MCV1 in some countries recommended in the second year of life, and more importantly, most countries now include MCV2 as integral part of routine immunization.
· Line 120. Spell-out WUENIC and may add link to website where it is explained
· Truly zero dose subsection – the caveat of MCV1 recommended in the second year of life should be made as children in the 12-23 m old group may be censored too early to have had enough time to receive the dose. In Latin America for example it is not that uncommon to end-up only with a MCV1 received when the child is aged 1 year.
· Variable Selection. May want to consider referring to this summary analysis https://www.who.int/publications/i/item/9789241511735
· 2.5. Estimating the zero dose and misclassified populations. Consider using the standard wording of “surviving infants”. Also, why was the number of estimated surviving infants derived rather than taken directly from the UN Pop Division database? It is available and it is what WUENIC uses.
Results [Besides changing the word “global” everywhere].
· Table 2. [Consider adding what % (or relative %) the misclassified would represent.
· Include the N number of countries in each income category, as the % alone can be misleading. This should be added to table 3.
· Birth order was not included and may be a confounder for some of the symptoms and access to care. If possible to add, it could be helpful. If not, probably should be added to the limitations.
· Table 7. Really add footnote on where the analyses don’t apply. I come back to BCG not making sense in Suriname as an example.
Discussion
Lines 377-379. Set of LMICs really. Unclear what “nationally” refers to here.
First/second paragraphs. Consider reviewing and citing recent papers on zero-dose by the Southampton group; and Gavi including Gupta & Hogan on ZD in April 2023.
Lines 431-433. Consider reviewing and citing recent papers on measles un-vaccination and how it relates to DTP ZD, I believe from the Pelotas group led by Cesar Victora.
Line 438. Unclear to me what “of popular secondary data” means.
Discuss further how to interpret campaign doses vs. routine and what is known re-overlap between people who receive campaign doses and are accessing routine immunization. Literature exists for polio and to a lesser extent for measles.
Limitations. Add that misclassification of immunization status is possible- see Dansereau et al, 2020 https://pubmed.ncbi.nlm.nih.gov/32270134/ , plus other misclassifications in DHS/MICS variables used, and the implications. The positive predictive value of a mother/caregiver saying “never vaccinated” may be higher than anything else. Even more accurate than no DTP.
Author Response
Dear Reviewer,
We greatly appreciate your thoughtful and detailed review. It has helped to improve the quality of our paper and its potential value to readers. We have made several edits to respond to your comments, as outlined below.
To improve the clarity of the work, as suggested, we changed the word ‘global’ to ‘overall’ throughout, since our analysis did not include all countries. In addition, we added a table listing the 82 countries in our analysis, the survey used in our analysis, their income grouping in the year of the survey, and notes about their vaccine schedule. In addition, we added the suggested references and agree that these have made our work stronger. Thank you for these thoughtful suggestions.
As suggested, we added paragraphs to better outline the advantages and limitations of administrative and survey data, as we agree that this is important context for our analysis. In addition, we clarified that we considered both campaign and routine immunization doses in our analysis, and that future analyses should disaggregate by delivery platform to improve strategy development.
We greatly appreciated the suggestion to revise our approach to estimate the population of truly zero dose, penta-zero dose, and misclassified zero dose children in the 82 countries in our analysis. We used the approach proposed by the reviewer that utilizes the data provided by WUENIC, to be aligned with WHO and UNICEF’s approaches.
We recognize the concern over variation in national immunization schedules and had also discussed this at length when designing our work. To clearly recognize that immunization schedules differ by countries, we included details on the country-specific immunization schedules in the table we added, and referenced this table where national schedule knowledge was relevant to interpretation of our results. As the reviewer noted, Suriname does not recommend BCG, which we now noted in our paper. This was the only country in our analysis that does not recommend BCG. In Suriname, a child who is truly zero would have a ‘0’ for DTP, polio, and MCV; additionally, all children in Suriname have a ‘0’ for BCG as it was not recommended to receive, and this would not change the subpopulation of truly zero dose children in Suriname. In our analysis, we are looking for the vaccine dose that would ‘push’ a child from being categorized as zero dose vs. not zero dose/vaccinated, so BCG (or any vaccine not recommended) would not influence the classification of zero dose. We recognize, though, that if were look at full immunization coverage (which we don’t do in this analysis), considering the national schedule would be necessary.
Since we aren’t looking at full immunization coverage, we do not look at the second doses of vaccines in our analysis, as these children would have received the first dose so would not be zero dose. There is therefore no need to look to MCV2, and the timing of this dose in the second year of life does not influence our analysis. Per the SAGE recommendations, all children should have received the first dose of the MCV at 12 months at the latest, with most countries in our analysis recommending MCV1 at or before 9 months.
We recognize the point on censoring and discussed this internally when designing this analysis. Since we look at children between 12 to 23 months of age, we expect that the children in our analysis have received the four vaccines. Since MCV is the last vaccine a child would receive in infancy, we expected that is unlikely that they would not receive BCG, DTP, and polio vaccines, but receive only MCV and would therefore not be truly zero dose. In exploring this in Table 7, we see that only 4.6% of children received only MCV and were misclassified as truly zero dose, so do not expect them the findings of our analyses to change substantially if we considered censoring differently. In addition, this four-vaccine approach is used by other researchers utilizing DHS/MICS data to assess zero dose status in children 12-23 months, as referenced in our work, so we think that using a similar approach lends comparability.
To address the point on misclassification of immunization status, we added text in the Methods and Discussion to clarify how data are collected in the DHS and MICS, our rationale for including both data on vaccine cards and caregiver recall, work that guided our approach, and the limitations of including caregiver recall data. We are looking at whether vaccinated children are misclassified as zero dose by looking at only DTP1 receipt, but it is not within the scope of our analysis to assess the quality of immunization data to accurately report immunization status.
While we considered the point to change the term from ‘penta-zero dose’ to ‘DTP-zero dose’, we decided to keep the term ‘penta’, but explicitly stated the countries that did not use the pentavalent vaccine. Of the countries in our analysis, 76 of the 82 countries have the pentavalent vaccine in their national schedule. In addition, we think that the term penta is used commonly among immunization program managers and researchers, so we feel that this is the appropriate term.
Lastly, we included in our limitations that we did not include birth order, and clarified our rationale for including variables that we expected to be associated with zero dose status, as well as misclassification of zero dose status. In addition, we incorporated the minor edits and suggestions by the reviewer (i.e. rephrasing text) to improve the interpretation and readability, and are grateful for the care the reviewer took in proffering these edits.
We are once again grateful for the reviewer’s thoughtful and insightful comments and suggestions. We are excited to resubmit this version of the manuscript.
Reviewer 2 Report
The manuscript provides an interesting approach, but it is very hard to read and this makes it less likely to widely used. The authors need to substantially tune down the complexity, provide better explanations of many topics and explain in simple terms what the study adds. The title is confusing – it should not be who is the zero child, but how to define the zero child. Namely, asking who implies finding the exact identity, which is not the aim of this study. Perhaps you need to define zero, truly zero and penta in the Asbtract, it will be more understandable. Table 1 – how do you reach 1/100%? Per row, column or table? Neither seems to fit. Table 2 – how did you inflate the sample size beyond your sample size? Table 3 – sums do not match 1/100%? Table 4 looks like an afterthought. Why would you consider this, since there are loads of possible confounding effects? Alpha is not equal to 0.05 but less than. Needs to be reported once, not at every instance. I think you did not define zero children, but stratified this by factors, what you are reporting in the Discussion.
Fine
Author Response
Dear Reviewer,
We are very grateful for your time in reviewing our manuscript. Your insight has helped us greatly to improve our paper. As you suggested, we revised the writing to make it more straightforward and less complex. We also edited the title, as we agreed with your comment about focusing on defining, not identifying, the zero dose child in the title. In addition, we include a definition of truly zero dose, penta zero dose, and misclassified zero dose in the abstract.
Thank you also for your comments on the tables. We included edits and clearer explanations to improve the interpretation of the tables. On Table 1 and 3, we do not aim to reach 1/100%. The columns here do not show all possible outcomes, so are not meant to sum to 100%. In Table 1, the rows shows the proportion of the outcome (penta-zero dose, truly zero dose, or misclassified zero dose), as well as the percentage of penta-zero dose children that are misclassified as zero dose; children who are not considered zero dose by any definition are not captured here, so the totals would not sum to 100% of the population. Similarly in Table 3, each row shows the proportion of the characteristic (i.e., poorest wealth quintile) among children described by each column. Since we again are only showing the prevalence of a specific characteristic among the population specified in the column, we do not expect this to each 100%. In Table 2, we did not inflate the sample size, but rather used population data to estimate the number of children in the countries in our analysis (not the number of observations) who are penta-zero dose, truly zero dose, and misclassified as zero dose; this is to provide an understanding of how many children the that the observations in the DHS/MICS represent. We did change our approach for focusing this to make it more straightforward, per another reviewer’s suggestion, and this is captured in the Methods. We edited the title of the table to clarify this. For Table 4, we recognize the potential confounders and added text in the Discussion to explain this limitation. In addition, we added an explanation of our rationale for including the analysis in Table 4, noting that we wanted to understand how child illness, careseeking, and connection may differ across the population of children prioritized in each zero dose definition, which may have programmatic implications of how zero dose children are targeted. In addition, we clarified that alpha<0.05 (not equal, as incorrectly stated). We kept this as a footnote in each table that had bolded results, to clearly denote what we were reporting and provide accurate interpretation of our results, but did not include it in other parts of the text, beyond when it was first stated in the Methods. In addition, we revised text throughout the discussion to make the interpretation of our analyses clearer.
We thank you again for the time and care you took in reviewing our analysis.
Reviewer 3 Report
This is an interesting analysis focusing on a key issue for immunization programs – that of children who miss out on vaccination. It makes use of the considerable data now available from multiple national surveys of vaccination coverage. The analysis uses fairly straightforward statistical methods, and comparisons. The paper is however, not easy to read through, and requires considerable attention to identify from the tables the data referred to in the text. I have suggested some edits which might make this easier.
Abstract : Date of surveys not included – refers to ‘most recent’ but would be useful to provide dates.
Line 21: ‘truly’ zero dose definition is unclear as to whether receipt of any one or combination of the listed vaccines less than all would still qualify as ‘truly zero’. I think what is meant is non receipt of any of the listed vaccines. Suggest add ‘any of’ after ‘received’ in line 21.
Introduction
Line 96-97 Provides a statement of the problem, but no clear statement of what the study aims to do / contribute. It would be useful to provide a statement of what the current analysis aims to contribute to this problem.
Materials and methods
2.1 Data. The use of two different survey sources for the data raises the question of whether data from the two surveys is sufficiently compatible that the data can be combined. My understanding is that the two surveys use very similar methodologies, in particular around measurement of vaccine coverage, so that the data can be combined. However, it would be useful to clarify this in the text.
Line 120 WUENIC data – abbreviation not spelt out.
Line 148-149: ‘have received at least one dose of the following vaccines’ implies that membership of this group requires receipt of one dose of each of the vaccines, rather than one dose of any one of the vaccines. Suggest ‘one dose of at least one of the following vaccines’.
Line 166-175: While the vaccination estimates refer to children aged between 12 and 23 months, the population estimates are based on estimates of the number of children surviving to age 12 months using birth data and estimates of infant mortality. What is the rationale for calculating the denominator populations based on the estimated population at age 12 months, when the definitions of penta-zero and truly-zero refer to the denominator population of children 12 to 23 months ?
Could use of this method also result in falsely high coverage estimates for countries with high rates of mortality in the second year of life ?
Results
Line 241 ‘haven’ should be ‘have’
Line 315-16 Would be useful to refer to Table 5 here to aid the reader and to provide definitions of overall vulnerability, and health access vulnerability in the text. It would also be helpful to clarify in tables 5,6 and 7 the category of the figures in the columns (population in millions, percentages etc)
Discussion and Conclusion
A useful comparison of the strengths and limitations of the analysis, with the conclusion providing a useful summary of the practical implications of the distinction between truly zero dose and misclassified zero dose groups.
Author Response
Dear Reviewer,
Thank you for your thoughtful and thorough review of our manuscript. We are encouraged to hear that you found the work to be interesting. We appreciate the time you took in reviewing our analysis; your insight and suggestions helped strengthen our work. We took several steps to improve the readability and clarity of our paper.
At the end of the introduction, we added to the statement of our aims to explicitly outline the specific objectives of our analyses. In the Methods section, we added text to explain the similarities and differences in the DHS and MICS and how they can be used together.
In addition, we are grateful for your feedback on the approach to estimate the population and this led us to revise our approach. Instead, we utilized WUENIC data of the target number of infants to give the population estimates in each country. The estimates were similar to our earlier approach, but we feel this approach is more straightforward and limits potential for bias.
We also clarified that our definition of zero dose includes ‘one dose of at least one of’ the four vaccines, as the reviewer suggested. In the Results section, we edited Tables 5, 5, and 7 to be clear that we were talking about proportions.
In both the Results and Discussion, we improved how we spoke about vulnerability, by adding the definitions of each vulnerability index used and the programmatic implications for the vulnerability analyses. Lastly, we made minor edits to improve clarity (i.e., spell out acronyms, fix typos, reference tables in the text, etc.).
Thank you, again, for your work in reviewing our analysis and helping us to improve our work. We are excited to resubmit this work.
Reviewer 4 Report
I appreciate the opportunity to critically review the article titled "Who is a zero dose child? Comparing different approaches to define zero dose children and how their profile changes based on the definition through an analysis of 82 low- and middle-income countries" In it, the authors present a very interesting analysis related to the definition of zero dose children. As a public health professional, I thoroughly enjoyed reading the article.
The document is clear and also presents an interesting proposal of two measures of vulnerability. I only have a minor comment for the authors:
Your findings revealed distinct vulnerability patterns when comparing truly zero dose children to misclassified zero dose children based on country income groupings. Specifically, starting from line 408, it would be useful for the readers if the authors could specify which vulnerabilities related to access to healthcare services they are referring to.
I would like to congratulate the authors on this highly relevant research work.
Author Response
Dear Reviewer,
I’d like to thank you for your supportive review of our analysis. We are encouraged to hear that you found the work to be highly relevant. We agreed with your point to improve the explanation of our vulnerability analysis. In the Discussion, we improved how we spoke about vulnerability, by adding the definitions of each vulnerability index used and the implications each index.
We thank you again for your review and appreciate your time in reading our work.
Round 2
Reviewer 1 Report
Thank you for considering the suggestions. I found annex A particularly informative and useful to determine where to investigate more in our own work.
My currents comments are minor:
Consider adding a footnote in the annex A table to countries where MCV is recommended at 12 m or more for completeness. I am being probably obsessive, but I think it would help the readers wanting to use the table (like me when I share this paper and call attention to the measles only, being <5% overall, but with high variability)
Annex A - spell-out DRC or add a footnote with country name, same for "PDR" in Lao PDR.
Correct typos in lines 450 and 508, as some typos occurred when the text was revised
Ref 6, separate Alliance from 2021
Ref 21, remove extra period after "...datasets]"
Ref 33, Use "World Health Organization" rather than "Organization WH" to be consistent with other WHO references
Ref 46, remove the duplicate "2021".
Finally, congratulations on your very clear, relevant and timely paper.
Author Response
Dear Reviewer,
Thank you very much for your continued support in our manuscript. We agree that the inclusion of details on the measles vaccine schedule is important, as this can help to interpret the findings that a small percentage of children receive only MCV. We have made the other improvements that you suggested. Thank you, again, for your time and insight.
Reviewer 2 Report
Thank you for the improvements, the manuscript now reads much easier
Author Response
Dear Reviewer,
Thank you very much for your review and your support in our manuscript. We are glad the manuscript reads better now.